# Degrowth Perspective for Sustainability in Built Environments

**Iana Nesterova** 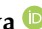

Department of Geography, Umeå University, 901 87 Umeå, Sweden; iana.nesterova@umu.se

**Definition:** Degrowth, as a social movement, a political project, and an academic paradigm, aims to find ways that can lead to harmonious co-existence between humanity and nature, between humans and non-humans, and within humanity, including oneself. Seen through the lens of degrowth, everything becomes subject to reflection, critique, re-evaluation, and re-imagining. This concerns environments created by humans in a long process of interaction with nature, i.e., built environments. Built environments are always in becoming. This entry contemplates the implications of degrowth for intentionally directing this becoming towards genuine sustainability.

**Keywords:** degrowth; post-growth; built environment; sustainability transformations

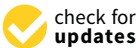



## 1. Introduction

Degrowth does not have a single definition, nor is degrowth a single discipline. It is at once a social movement, a political project, and an academic paradigm [1]. What unites these pursuits is a desire for sustainable, harmonious co-existence between humanity and nature, between humans and non-humans, and within humanity, including oneself [2,3]. The name of the concept, degrowth, reflects the roots of the concept, i.e., a critique of prevailing economic growth orientation which disregards the negative effects of pursuing economic growth, such as ecological and social degradation [2,4]. Despite its name, degrowth is not opposed to growth, but the growth that it advocates, supports and encourages is mainly non-material, such as growth in care, solidarity, empathy, creativity, generosity, and connectedness. Neither is material growth fully alien to degrowth thought. Selective growth (such as in renewable energy provision, in organic agriculture and permaculture) is welcomed, as are improvements in material conditions of those whose genuine needs are not met. While degrowth originated in the 1970s with the rise in prominence of a variety of ecological movements, the ideas which degrowth scholars explore originated much earlier and have longer intellectual heritage and histories. For instance, degrowth calls for simpler living, deviation from consumerism, and viewing non-humans as one's neighbours, calls that could be readily found in the 19th century (see, e.g., [5]). Likewise, in the 19th century, the "hunger for wealth, which reduces the planet to a garden", and the need to live harmoniously in and with nature, were noticed [6] (p. 161). Going even further back in time, harmonious co-existence with nature, as well as with other humans, were prominent thoughts in ancient China [7].

As an academic paradigm, degrowth is very broad. While influential scientists within this paradigm can be identified [1], including perhaps most notably Nicholas Georgescu-Roegen [8,9] and Serge Latouche [10], currently, multiple scholars from a wide variety of disciplines are working with the concept of degrowth which contributes to the diversity within this academic paradigm. They include sociologists (see, e.g., [11]), political economists (see, e.g., [12]), and geographers (see, e.g., [13]). In recent years, adventurous inter-disciplinarity has become even more evident in the broad and growing field of degrowth. For instance, degrowth has been included in dialogues with philosophies such as existentialism [3] and critical realism [14], and with diverse economies thinking [15]. Degrowth thinking has been applied to multiple phenomena, such as business and organisation [3,13,15–17], housing [18], tourism [19], and technology [20]. Many other

strands within academia share similar pursuits with degrowth, exemplified in a desire for a good and harmonious life for all, including humans, nature and non-human beings. Such strands include, for instance, diverse economies [21], deep ecology [22] and techno-logical scepticism and pessimism [23], which are often found in dialogues with degrowth. Outside academia, a variety of movements and initiatives can be seen as degrowth compat-ible, whether they explicitly identify themselves as such or not [24]. Implicitly degrowth compatible movements such as voluntary simplicity, the tiny house movement and the zero-waste movement share many commonalities with degrowth but do not necessarily identify themselves as part of degrowth, though some practitioners might. As a political project, degrowth is likewise diverse. In terms of political ideologies, some see degrowth as an eco–socialist project [25], while others advocate for anarchism [26] and call for bottom-up transformations. Such transformations entail, first and foremost, changes in individuals' values and worldviews, and progress in moral agency rather than (high) technology [3,27]. Yet others do not propose a single political ideology but rather see the current political systems as diverse. They propose a variety of pathways of how degrowth can be achieved, which includes both top-down and bottom-up strategies [12]. The journey of degrowth as a social movement, a political project and an academic paradigm is unfolding, and the collection of knowledge is becoming ever deeper, more diverse, and broader. Since degrowth seeks answers to the question of how a harmonious co-existence can be achieved, nothing remains unquestioned. Every social domain, phenomenon, norm, institution can be looked at from a critical degrowth perspective. Likewise, the deep and broad knowl-edge that degrowth has accumulated so far can be used for proposing better alternatives that ensure a good life for all humans and non-humans far into the future. This includes built environments.

Built environments are environments that humans have created in a long process of interaction with nature, claiming nature's own spaces and transforming them into their own. Such human spaces can be viewed as "cocoons that humans have woven in order to feel at home in nature" [28] (p. 13). Such cocoons span across the face of the earth, affecting even those areas where human presence is not immediately visible. Untouched natural environments that, for instance, von Humboldt [29] describes, largely remain in the past. Even where no human activity is visible to the naked eye, the effects are felt due to human imposed activities resulting in climate change, ocean acidification, pollution, and other detrimental effects.

Built environments are diverse, ranging from simple dwellings to large and complex cities [28]. They depend on cultures, geographies, and topographies, and are always in de-velopment. They are constantly unfolding and changing. Thus, the past and past participle form of the verb, as well as the adjective, "built" gives a false sense of completion, stability, and finality. In critical realist terms, reality consists of social and physical realities [30]. Built environments are where social and physical realities meet, empower, and impose constraints onto each other. A question arises: what could be implications of degrowth thinking for sustainability in built environments?

## 2. Degrowth in Built Environments

While it has become normalised to see humans and nature as opposites, ontologically this is not the case [31]. There is no strict separation between humans and nature. Humans are embodied personalities, and one's body is of nature. In its turn, nature is impacted by humans, our pursuits and creative thought manifested in action and transformation. Nature shapes humans' worldviews [22,28] and has always provided inspiration for humans, including in its own transformation [32]. Built environments provide excellent examples of inseparability between humans and nature. For instance, a house is made from materials derived from nature, assembled together (and most often collectively) by humans guided by their creativity and skill. Looking closer at the house, one may notice ladybirds on a balcony and birds residing under the roof, building a nest. The house is home not only for humans, but also for non-humans. Due to this inseparability, it is perhaps best to see ourselves not as

tenants in a particular house or a town, but rather as "tenants of the earth" [33] (p. 7). This is precisely what degrowth scholars advocate and encourage humans to do. Beyond seeing ourselves as tenants of the earth, it is essential to behave as such and in practice strive for sustainable, harmonious co-existence within humanity and with nature. The implications of this thinking for built environments are profound, though, as we will see, not always precise at this stage of degrowth's journey as an academic paradigm.

From a degrowth perspective, the central question when contemplating built environments is the question of harmonious co-existence. It is perhaps best captured by Thoreau's [5] chapter title in Walden, "Where I lived, and what I lived for", but this time applied to societal level as well as the level of a human individual. Degrowth does not provide a concrete answer to the question of what a human individual or societies should live for. However, it does make suggestions regarding what is destructive and may be misleading. While it calls for a deviation from materially excessive, hedonistic lifestyles, it invites humans to focus on justice, an increase in local self-sufficiency, independence from global economic forces and from commercialisation [34–36]. It encourages humans to make a step from overproduction and overconsumption to sufficient production and consumption, from material wealth to wellbeing [2]. In this invitation lie hints and more precise suggestions of where to live, how to approach our built environments, how to make them more sustainable.

It is suggested that in a degrowth society, community needs to replace commerce [37]. Thus, the life of human settlements will not be structured around commercial pursuits, commercial infrastructure, daily migration for work, shopping, and other market centred activities. It will be centred around community and fully exploring the possibilities of being in the world with others [38]. The focus of degrowth living is primarily on locations, i.e., the town, the suburb, the neighbourhood where most needs will be met [36]. Trainer [36] paints a picture of what a degrowth built environment may look like and entail in practice. It would include backyard and communal gardens and workshops, and multiple small, privately and collectively owned businesses instead of large businesses. Travelling on foot or by bicycle would replace travelling by private cars [36], thus creating a need for bicycle compatible infrastructure. Railway services would remain and, in most cases, replace air travel, which naturally leaves us with questions regarding existing airports and associated with them infrastructure. Trainer [36] suggests that derelict sites would be repurposed for productive activities by communities, and streets would become permaculture commons. Existing retail infrastructure could be repurposed to house repair stores or become meeting places, libraries, and places of barter [36]. Other suggestions comparable to Trainer's [36] include ecovillages [39] and intentional communities, which challenge the norm of what built environment in industrialised countries should look like and be centred around. They actively deviate from standardisation that is common in capitalist settings [4]. While such visions may appear utopian and radically different from what our environments currently look like, such alternatives exist already, though in the niches and pockets of current societies [21]. In a degrowth society, such initiatives would come to prominence and claim spaces. Such claiming of space should be seen as a process rather than a sudden shift. Oftentimes, existing alternatives and modes of being have one foot in a capitalist economy and the other in a non-capitalist economy [40], making the process of transformation and space claiming challenging and arduous, though not impossible [41].

What characterises degrowth inspired thought in relation to built environments is diversity, plurality, possibility of change and alternative modes of living, relating and beings. Diversity, for instance, can be seen in the forms of housing for degrowth (tiny house, yurts, using natural, reclaimed, repurposed materials), as well as in living arrangements (e.g., co-living) and ownership possibilities (common rather than private). While in the current capitalist society the emphasis is put on private property and the isolated individual, degrowth emphasises togetherness, transcending the situation of ecological and social degradation in cooperation with others.

Being with others includes fellow humans, but also non-human beings [3]. Thus, the notion of the built environment itself from a degrowth perspective needs to be infused with a different and much expanded meaning. It needs to recognise labour beyond human labour [42] in building, such as the labour of nature itself in providing materials for built environments and keeping these environments pleasant, healthy, and liveable. It also remains important to acknowledge the built environments of other species, such as beehives, termite mounds, rabbits' burrows, and ant nests, where ants live in symbiosis with fungus and establishing fungus gardens [42]. Human beings are not the only beings who transform their environments, and harmonious co-existence with others presupposes respect for another being's built environment, dwelling and space. Moreover, non-human beings such as birds, foxes, bees, and hedgehogs live in proximity to and within built human spaces such as cities, gardens, and parks. Since degrowth emphasises the wellbeing of others beyond humans, transformations of human-built environments need to be attentive to the diversity of beings within human-built spaces, ensuring their safety and allowing them to flourish.

Apart from directing our imagination towards more harmonious built environments, degrowth offers useful suggestions which can assist transformation in practice. For instance, creating new spaces should focus on nurturing healthy relationships with nature, others, and the self. Tuan [28] notes how different places interact with humans' senses differently. While a skyscraper is a pathological space, "the experience of the interior of a cathedral involves sight, sound, touch, and smell. Each sense reinforces the other so that together they clarify the structure and substance of the entire building, revealing its essential character" [28] (p. 11). One's experience in a forest can be even more profound in terms of connection and self-transcendence [5,32]. New buildings can be inspired by non-commercial spaces and by nature and use natural, renewable, biodegradable materials. Wherever possible, repurposing existing buildings and infrastructure should take place [36]. Reduction in resource use is central to degrowth, thus making use of the existing stock should be considered, and improvements (insulation, increased energy efficiency, water reuse systems, safety) made where possible. Degrowth scholars often question the pursuit of constant technological innovation [10,20], thus less technologically intensive solutions can be sought. Such solutions can be sought outside Western science: degrowth scholars emphasise plurality of knowledge, such as lay and indigenous knowledge [43]. Another proposal is removing advertisements from human spaces which take attention and direct it towards consumption and having rather than being [44].

While such a wide diversity of considerations may seem overwhelming, it is possible to structure them around a limited number of domains or planes of social being. For instance, Bhaskar [45] suggests that social being, and any social phenomenon, exist and unfold on four interconnected planes. They include (1) material transactions with nature, (2) social interactions, relationships between people, (3) social structure, and (4) embodied personality. When considering transformations of built environments or any aspect thereof, one may consider, and become attentive to all of these planes at once. For instance, when planning a housing development, in terms of material transactions with nature, one may consider the geography of the location and the materials to be used, whether they are biodegradable, durable and will serve a long time, can be sourced locally or reused. In terms of the second plane, one may consider the possibilities of communal activities which would facilitate healthy human interactions, nurturing relationships and cooperation, such as establishing and looking after a community garden nearby. The third plane, that of social structure, is abstract and unites a plurality of social structures, such as norms, institutions, as well as buildings themselves. On this plane, one may consider a wide variety of aspects such as affordability of housing, accessibility and quality of useful infrastructure, and challenging the norms (e.g., deviation from standardised architecture, seeking creative architectural solutions, changing attitudes towards non-humans). The plane of embodied personality is local and intimate and concerns oneself. While degrowth tends to overlook this dimension, the self is essential to consider, since this is where creative, transformative

change resides [46]. On this plane, one may consider whether a house may reflect the personality and the needs of a human being, if it can become a place and not only a location to a person, if it accounts for the complexity of human nature (such as the drives towards and also away from other humans).

Recently, the scientific pursuits of degrowth scholars were referred to as a science of deep transformations [14]. Deep transformations concern not only scientific pursuits, but also, and perhaps most importantly, the nature of transformations in the real world. From this perspective, a built environment needs to be deeply transformed and become deeply transformative for human beings; it needs to produce spaces for sustainable, harmonious, peaceful co-existence, for humans and non-humans, for life (e.g., homes) and also death (e.g., cemeteries).

## 3. Constraints

While spaces are socially created [47] and thus, like all social structures, are only relatively enduring [41], there are constraints to the unfolding of degrowth compatible built environments. Despite the growth in degrowth as a movement, a political project and an academic paradigm, degrowth still does not enjoy wide support from the general public [48]. While degrowth indeed pursues human flourishing, its emphasis on nature and non-humans, whose flourishing degrowth likewise defends, may invoke scepticism from the electorate [49]. Thus, attempts to reclaim spaces and repurpose them, e.g., replacing car parks with gardens, may not be welcomed by the public. A change in values and worldviews is required for such proposals to be met with a positive attitude [3]. A wide range of constraints is related to capitalism as a system, including state capitalism. Capitalism is a powerful hurdle on the path of transformation [4,15,50]. This is because this system is orientated towards valorisation of capital [50]. It is evident, for instance, in the strife of businesses to make a profit [16,51] and more generally, in expecting monetary return on one's investments. Capitalism is centred around private property and often its accumulation and rent, thus ownership rights. Ownership rights need to be modified for degrowth built environments to be possible, so choices can be made not on the grounds of what is economically best for the owner (e.g., to build a shopping centre, a standardised block of flats, an airport, or a mansion), but what is best for the sustainable thriving of human and non-human life (e.g., an affordable housing development, an organic agriculture project, or a forest). However, with ownership rights come responsibilities. These include responsibilities for health, safety, and maintenance of built environments. If ownership rights are transferred to communities, questions concerning these aspects and how decisions can be made democratically will need to be considered.

## 4. Conclusions

This entry attempted to capture the implications of degrowth for built environments. First and foremost, an important implication is ontological, i.e., being mindful, as scholars, practitioners and citizens of the ontology of human societies. This is exemplified in the acute realisation of our inter-connectedness with nature. Every human act of transformation in our built environments and beyond entails the use of natural resources and energy. Due to the ongoing ecological degradation, such transformation needs to reduce as much as possible.

Furthermore, the design, building and repurposing of sites, from a degrowth perspective, would be aimed at much simpler, place-sensitive and more local and communal living. Thus, principles such as simplicity, localisation and community can be used to guide urban and beyond-urban planning. The question of localisation, local built environment and local living is of immediate relevance. At the time of writing this manuscript, the world is facing the COVID-19 pandemic. As one of its implications, many humans found their lives constrained to their local areas. While neither this crisis itself, nor the sudden localisation of living can be described in terms of degrowth, the localisation of living showed the importance of the local area, local community, local infrastructure and local nature. While

some found themselves in attractive locations in proximity to natural settings and useful infrastructure, others found themselves in areas where the traces of natural environments are scarce, the skyline is human-made, and where tall buildings conceal the horizon. It is challenging to know what role cities may play in a degrowth society. However, degrowth may provide inspiration for better localised living. Creation of accessible green spaces, encouraging community life and solidarity, proliferation of local small businesses, creating opportunities for outdoor exercise, spiritual practices and nature connection are but some elements of the direction in which degrowth inspires us to go. This requires a profound change in values, since pursuing this direction entails deviation from the opposite: appropriation of green spaces, atomised existence, car-centred infrastructure, shopping centres and other manifestations of modern built environments. Since degrowth aims at a good life for all, what remains essential to consider is accessibility to healthy and harmonious built environments for everyone, not only those who can currently afford it.

A useful way to apply degrowth to built environments in order to make them more sustainable is to think in terms of four planes of social being at once, which captures all aspects of social ontology, including humans' material transactions with nature, social relationships, social structures, and the self. While degrowth provides a wealth of ideas and encouragement to think freely, creatively, adventurously and to embrace alternatives, there are constraints to a degrowth compatible built environment. Apart from the capitalist system itself, which has valorisation of capital and not sustaining life at its core, there is a variety of its manifestations which act as constraining structures. They include, for instance, profit seeking and ownership rights. Despite these constraints, alternatives thrive currently and are plentiful, and the built environment will without a doubt also be part of degrowth society. Opportunities for contemplating what it will look like, and bringing harmonious and nurturing alternatives to existence, remain immense.

**Funding:** This research received no external funding.

**Data Availability Statement:** Not applicable.

**Conflicts of Interest:** The author declares no conflict of interest.

**Entry Link on the Encyclopedia Platform:** https://encyclopedia.pub/20570.

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
