# Peer review of "Degrowth Perspective for Sustainability in Built Environments"

_encyclopedia, doi:10.3390/encyclopedia2010029_

Round 1
Reviewer 1 Report
There is some ideological bias in the way capitalism is presented in relation to degrowth.
Indeed, degrowth is affected by a range of constraints that have not been tackled by this study, thus rendering this part incomplete.
The style of writing is simplistic and fails to endow the text with the coherence levels one would expect from such a contribution.
The conclusions are somewhat generic, speculative and devoid of any pertinent recommendations.
Reviewer 2 Report
This is a very pertinent entry that should be published. However, I think it could be improved if the author refers:
- The problematic relationship between progressivism and degrowth.
- The thought of Bruno Latour and his influence.
Reviewer 3 Report
This is a very good text. Some comments and sugestions.
As this is an encyclopedic text - consider entering a short encyclopedic password, possibly with a colon's padding, e.g. https://doi.org/10.3390/encyclopedia2010013
sugestion (?): Degrowth in built environments: a sustainable approach
It seems to me that the abstract does not fully describe what is in the text. I think it is worth rewriting it to meet the standards of encyclopedia.
Conclusions are a bit too long. Some of the material is repeated. I think choosing the actual conclusions would help to summarize the text.
General remark - some sentences are too long - I advise you to shorten them, as you often do not know what the main idea is.
Round 2
Reviewer 3 Report
I am happy with the current version.